# A Focus Group Study about Oral Drug Administration Practices at Hospital Wards—Aspects to Consider in Drug Development of Age-Appropriate Formulations for Children

**DOI:** 10.3390/pharmaceutics12020109

**Published:** 2020-01-30

**Authors:** Maria Rautamo, Kirsi Kvarnström, Mia Sivén, Marja Airaksinen, Pekka Lahdenne, Niklas Sandler

**Affiliations:** 1HUS Pharmacy, HUS Helsinki University Hospital, Stenbäckinkatu 9B, 00290 Helsinki, Finland; kirsi.kvarnstrom@hus.fi; 2Faculty of Pharmacy, University of Helsinki, Viikinkaari 5E, University of Helsinki, 00014 Helsinki, Finland; mia.siven@helsinki.fi (M.S.); marja.airaksinen@helsinki.fi (M.A.); 3Pharmaceutical Sciences Laboratory, Åbo Akademi University, Tykistökatu 6A, 20520 Turku, Finland; niklas.o.sandler@gmail.com; 4Department of Children and Adolescents, HUS Helsinki University Hospital, Stenbäckinkatu 9, 00029 Helsinki, Finland; pekka.lahdenne@hus.fi

**Keywords:** oral drug administration, drug administration challenges, qualitative study, pediatric, drug manipulation, age-appropriate

## Abstract

Oral drug administration to pediatric patients is characterized by a lack of age-appropriate drug products and the off-label use of medicines. However, drug administration practices at hospital wards is a scarcely studied subject. The aim of this study was to explore the oral drug administration practices at pediatric hospital wards, with a focus on experiences and challenges faced, methods used to mitigate existing problems, drug manipulation habits, perceptions about oral dosage forms and future needs of oral dosage forms for children. This was a qualitative study consisting of focus group discussions with physicians, nurses and clinical pharmacists in a tertiary university hospital with the objective of bringing forward a holistic view on this research topic. These healthcare professionals recognized different administration challenges that were classified as either dosage form-related or patient-related ones. A lack of depot formulations developed especially for children as well as oral pediatric dosage forms of drug substances currently available as intravenous dosage forms was recognized. The preferred oral dosage forms were oral liquids and orodispersible tablets. Patient-centered drug administration practices including factors facilitating drug administration both at hospital wards and at home after patient discharge were identified. Among all healthcare professionals, the efficient cooperation in drug prescribing and administration as well as in educating the child’s caregivers in correct administration techniques before discharge and improving the overall discharge process of patients was emphasized. This study complements the prevalent understanding that new dosage forms for children of varying ages and stages of development are still needed. It also brings a holistic view on different aspects of oral drug administration to pediatric patients and overall patient-centered drug administration practices.

## 1. Introduction

Oral administration to pediatric patients is characterized by a lack of age-appropriate drug products and the off-label use of medicines. This off-label use implies that the indication or route of administration differs from the summary of Product Characteristics (SPC), the use in pediatrics is contraindicated or that the age or weight of the patient is below the specifications in SPC. In a study conducted in Swedish hospitals, 60 percent of hospitalized children had at least one medicine prescribed off-label [1]. The off-label use of medicines was most common for neonates and infants. Recently, the off-label use of medicines to children has been addressed on a European level as the European Academy of Pediatrics and the European Society for Developmental Perinatal and Pediatric Pharmacology published a statement on recommendations on the appropriate, rational and safe prescribing of off-label medicines to pediatric patients [2]. The lack of age-appropriate medicines is also generating the need to physically modify drug products. This manipulation concerns mainly solid oral dosage forms and is done to enable the administration of an appropriate dose or enhance medicine administration [3,4]. The typical methods for manipulating solid dosage forms are the splitting and crushing of tablets, opening of capsules and dispersing powder in liquid or food [4]. Drug administration through enteral feeding tubes is also related to the dispersion of solid dosage forms [5]. Recommendations considering the risk of physical blockage of the feeding tube as well as the chemical compatibility of a drug product with the tube material have been published by European Medicines Agency (EMA) in a guideline on pharmaceutical development of medicines for children [6]. Such events would affect the administered drug amount; however, potential drug loss can be shown by mimicking drug administration through enteral feeding tubes for children [5,7].

The ideal orally administered formulation for children would enable flexible dosing, be palatable and easy to administer, contain only child-safe excipients and be stable from microbiological, physical and chemical perspectives [8]. Flexible dosing is often linked to liquid dosage forms, multi-particulate formulations and mini-tablets, whereas solid dosage forms are associated with good stability [9]. A risk of exposure to excipients through the oral administration route exists, since excipients are used in oral dosage forms to improve, for instance, patient acceptability, the solubility of the drug substance or the shelf-life of the drug product [10]. Excipients such as preservatives, colorants, sweeteners, ethanol and propylene glycol can be harmful to children, especially for infants and neonates [10,11,12]. Garcia-Palop et al. discovered in a Spanish University Hospital that at least one harmful excipient was present in 62% of orally administered drugs used for neonates [13]. Further observations revealed that excipient amounts exceeded the recommendations for maximum daily intake for adults in 19% of oral medicines. Additionally, a Danish study showed that the exposure of benzyl alcohol, ethanol and propylene glycol in hospitalized neonates and infants less than 2 years of age was substantial and the cumulative daily exposure frequently exceeded the tolerance limits stated by EMA [14].

The age of pediatric patients was identified as the main aspect influencing the choice of dosage form [15]. An observational study reviewing oral drug administrations to pediatric inpatients aged 0–15 years over a one-year period showed that a liquid dosage form was used in more than half of all oral drug administrations [16]. Most often, liquid dosage forms were administered to children younger than 6 years of age and in more than 40% of drug administrations to schoolchildren (6–11 years). Solid dosage forms including tablets, capsules, orodispersible and chewable tablets were used in 28% of the total number of oral drug administrations to children. In children aged 6–17 years and their caregivers, Ranmal et al., using a questionnaire, assessed attitudes towards choice of six solid dosage forms (tablets, capsules, mini-tablets, chewable and orodispersible tablets and multiparticulates) [17]. This study evaluated opinions rather than experiences since the investigated dosage forms were not actually administered to the children, although some children had previously used other solid dosage forms except for mini-tablets. Among children aged 6–11 years and their caregivers, chewable tablets were the most favored solid dosage form followed by orodispersible tablets and mini-tablets whereas tablets and capsules were the least preferred dosage forms. Approximately 80% of adolescents aged 12–17 years found tablets to be the most acceptable dosage form, followed by chewables, orodispersible tablets and mini-tablets. Recent drug acceptability studies have shown that mini-tablets are well tolerated by pediatric patients since children aged 6 months to 6 years preferred one placebo-containing mini-tablet, having a diameter of 2mm, to glucose syrup [18]. Infants and preschool children were also able to swallow multiple mini-tablets as one dose [19].

Drug administration practices in hospital wards is a scarcely studied subject. In the past few years, other working groups have investigated children’s, their caregivers’ and healthcare professionals’ preferences for different oral dosage forms, formulation-related barriers to oral medicine administration in children as well as execution of oral administrations to hospitalized children [3,15,16,17,20]. To the knowledge of the authors, there is only one paper published on healthcare professionals’ opinions about problems related to prescribing, dispensing and administering oral formulations to children [21]. The aim of this study was to explore the oral drug administration practices at pediatric hospital wards, with focus on experiences and challenges faced, methods used to mitigate existing problems, drug manipulation habits, perceptions about oral dosage forms and future needs of oral dosage forms for children. The objective was to bring a holistic view to the research topic by carrying out this study in healthcare professionals in a tertiary hospital.

## 2. Methods 

The study was conducted as focus group interviews at a tertiary university hospital. The physicians, nurses and clinical pharmacists who were invited to participate in the study worked at the Department of Children and Adolescents, HUS Helsinki University Hospital. The department provides specialized health care for children ranging from neonates to 15-year-olds in all areas of pediatrics, pediatric surgery, child neurology and child psychiatry in its catchment area in Southern Finland. In addition, the department provides care for pediatric patients from across Finland in severe cardiac problems and organ transplantation, as well as other rare conditions requiring demanding tertiary care. HUS Pharmacy provides hospital pharmacy services and clinical pharmacists for the Department of Children and Adolescents.

### 2.1. Study Design and Data Collection

The qualitative design of focus group discussions was chosen because it is a useful method to explore the beliefs, behaviors and attitudes of individuals [22]. Themes can evolve both from the chosen research topics as well as from subjects emerging during the focus group discussions amongst the participants. In this study, we used an interview guide of semi-structured questions to moderate the discussion (Table 1). The interview guide was constructed to reflect the study aim and it was tested in a pilot interview. The interview guide was not altered following the pilot and the pilot interview was included in the research data. The principal investigator (MR) facilitated the interview sessions assisted by another investigator (KK). Both investigators have a background in hospital pharmacy.

We used purposive selection of participants in order to get a comprehensive representation of the different pediatric subspecialties. This type of selection is useful when participants ought to have something to say about a given topic [22]. Head Physicians of the different pediatric subspecialties recruited the physicians, a Nurse Director recruited the nurses and the principal investigator (MR) recruited the clinical pharmacists. The recruitment method was email invitation. The interviews took place between May and September 2018. Each focus group discussion consisted of participants from only one profession.

### 2.2. Qualitative Analysis

The focus group discussions were digitally audio-recorded and transcribed verbatim. Inductive content analysis was used to analyze the transcripts utilizing a framework analysis approach. First, all quotes were identified. The quotes were then coded and codes with a similar meaning were systematically rearranged into subcategories and categories using Microsoft Excel (Microsoft Corporation). The transcripts were separately coded and categorized by two investigators (MR and KK). Any differences in the analysis were discussed until a mutual opinion was reached.

### 2.3. Ethics and Informed Consent

The Ethics Committee of Helsinki University Hospital granted ethical approval (HUS/3637/2017). All participants received written information about the study before they gave their written informed consent.

## 3. Results 

A total of 19 participants were recruited and divided into five groups (Table 2). Each group consisted of 3–5 participants, all of them having the same profession (physician, nurse or pharmacist). The focus group discussions lasted for one hour.

Five different themes emerged from the focus group discussions during the coding of the transcripts (Figure 1). Drug administration challenges were further divided into two subcategories—dosage form-related and patient-related challenges.

### 3.1. Drug Administration Challenges

Physicians, nurses and clinical pharmacists reported many different administration challenges (Figure 2).

#### 3.1.1. Dosage Form-Related Administration Challenges

Tablets and capsules were considered difficult to administer to children. One problem is that the drug dose in the tablet or capsule is intended for adults, thus making them unsuitable for children without modifying the dosage form. Tablets containing multiple active pharmaceutical ingredients (API), such as medicines against HIV or tuberculosis, are particularly problematic because the required dose of each API in the combination is dependent on the child’s age.


*There are not such tablets available that we would need and then we have to cut or crush or look for other dosage forms.*


Depot tablets or capsules cannot be divided into smaller parts to get the right dose because the pharmacokinetics of the medicine would change. Thus, it is impossible to give depot formulations containing unsuitable doses to children even though the medical condition of a child would benefit from receiving the medicine as a slow release dosage form. Examples of such conditions are pain relief and treatment of epilepsy. The size of depot formulations is often so big that children cannot swallow them. Furthermore, it is impossible to crush and administer depot formulations to patients that have a nasogastric tube or percutaneous endoscopic gastrostomies.


*Younger children cannot swallow depot formulations. Depot formulations cannot be administered to older children withan enteral feeding tube.*


The volume of the dose can sometimes be too large for a child to swallow. Liquid formulations are easy to swallow in general, but if the amount of liquid exceeds a few milliliters, it can result in spitting or vomiting. The content of the API in the formulation should enable the administration of the product to different age groups without making the volume of the largest dose too big. An example of a problematic liquid formulation is oral solutions containing paracetamol. The dose volume often becomes too big even for infants.


*The volume has quite a big impact. If it’s small, it’s much easier to administer than if it’s 5 mL or 10 mL.*


Some drugs are available only as formulations for intravenous administration. However, there are situations when oral administration is the primary route of administration of intravenous drugs. For these drugs, a need for an orally administered dosage form is apparent. Ketamine, for the treatment of pain, vancomycin, midazolam and electrolytes are some examples of intravenous drugs also administered orally.


*We use the intravenous dosage form of midazolam because the concentration of the commercial oral formulation is so small that we would have to administer a large volume, which is not possible when a child has fasted.*


The taste and texture of a drug formulation are important features affecting drug acceptability. Taste issues occur especially when intravenous drugs are administered orally and for some liquid formulations. Orally dispersing tablets are challenging for some children because of the texture and mouth feeling.


*A tablet dispersing in the mouth always has a taste and a feeling that a mass remains in the mouth.*


Manipulation of dosage forms is widely used to get individual doses and manage swallowing difficulties. According to the healthcare professionals, techniques such as cutting and crushing of tablets, emptying of capsules, ex tempore manufacturing of dose powders and liquid formulations, as well as the dilution of liquid formulations, were applied on a regular basis. The interviews revealed a concern for dose accuracy related to some drug manipulation practices.


*However, if the volume is very small, you get uncertain if the child gets the whole dose and if there is some loss during the administration.*


All focus groups discussed formulation-derived administration challenges. Some patients have suffered from adverse drug events caused by excipients. A more common administration challenge is, however, the blockage of the enteral feeding tube because of cellulose-based excipients. The pharmacists considered ethanol, propylene glycol and parabens potentially harmful to children.

#### 3.1.2. Patient-Related Administration Challenges

Patient-related administration challenges emerging from the data are swallowing, spitting, vomiting, the need for an enteral feeding tube, age and individual taste preferences (Figure 2). Swallowing tablets can be challenging for some children regardless of age and size of the tablet. On the other hand, some children can learn to swallow tablets at a very young age, whereas some adolescents are not capable of doing it. If a tablet starts melting in the mouth and tastes bad, a normal reaction to this is spitting. Some infants tend to also spit liquid dosage forms if the volume is larger than 0.5 mL. For some children, the administration of tablets or big amounts of liquid formulations induces the gag reflex. The risk for this is more likely the more medicine the child has to swallow at one time or if the medicine tastes bad. Taste preferences are very individual.


*Some children automatically bite the medicine and start tasting it.*


Regarding the impact of the age of a child on drug administration challenges, newborns and infants are the most challenging groups of children. Infants are not accustomed to ingesting anything other than milk. Some newborns have not even had their first drink of milk before their first oral drug administration. The need for dose adjustments based on body weight is routine at pediatric wards. It is also common that these patient groups in particular have an enteral feeding tube.

Enteral feeding tubes place specific requirements on oral drug administration. All solid oral dosage forms have to be crushed and dispersed or diluted in fluid before administration. The enteral feeding tubes must be rinsed with water between administrations of different medicines. Neither the dispersed solid dosage forms nor the liquid oral dosage forms should cause clogging of the feeding tubes. If this happens, the tubes must be rinsed with further amounts of water. This can be crucial when individual fluid restrictions place limitations on the amount of liquid administered to a child. It is important to consider these requirements in drug development.


*The smaller the child, the smaller the tubes and the more easily they clog and then they have to be flushed and that is truly a significant problem.*


### 3.2. Suitable Dosage Forms for Pediatrics 

According to healthcare professionals involved in the focus group discussions, orodispersible and liquid dosage forms are suitable for most pediatric patients (Table 3). For infants, the liquid dosage form is easiest to administer. Another benefit is the suitability for dose adjustments, which are very common as the children grow. For example, ex tempore compounding of a cancer medicine has improved the adherence and thus the treatment outcomes. A disadvantage of liquid dosage forms is inferior shelf-life compared to solid dosage forms.

Orodispersible tablets are suitable to administer directly into the mouth or to be dispersed in a small amount of liquid before administration. Therefore, they were considered to be a good oral dosage form for children, starting from infants. Solid dosage forms such as tablets and capsules were only considered suitable for children who can swallow them whole. This ability was not considered to be connected with any specific age, but rather a completely individual capability.

### 3.3. Factors Promoting Successful Drug Administration at Hospital Wards

The expertise of nurses plays an important role in managing the successful delivery of oral drugs (Table 4). It is important to consider the preferences of an individual child when choosing the oral dosage form and adapting different manipulation techniques. Juices or juice concentrate, milk, fruit purees or jam are used to cover the bad taste of a drug. The possibility of limiting the total volume of liquid to be administered directly effects the willingness of the child to take the medication. A useful aid to facilitate the swallowing of tablets and capsules is covering them with a tasteful coating.

### 3.4. Factors Promoting Successful Drug Administration after Patient Discharge

Before patient discharge, a nurse or clinical pharmacist teaches the child’s caregivers how to administer drug products correctly. Useful drug manipulation techniques as well as drug administration through an enteral feeding tube, are important to manage completely. Caregivers of severely ill children are motivated to succeed with drug administration and are able to manage it quite well. However, there might be language barriers that complicate the education of caregivers.


*Sufficient instructions about the handling and administration of medicines must be given at patient discharge.*


The availability of uniform drug products and medical supplies in the hospital and at home facilitates patient discharge and the ability of parents to succeed in drug administration at home. At the moment, problems occur because both the drug products and medical supplies used in the hospital differ from the ones the family can buy at a community pharmacy after discharge. For instance, oral syringes are not available at community pharmacies. It is confusing for parents when the scale on the syringes differs from the ones used at the training situation. Generic substitution is also confusing for parents if the tablet bought at the pharmacy looks completely different from the one used at the hospital. The excipients can also differ, which might lead to differences in solubility and cutting difficulties. Some physicians forbid generic substitution to avoid these problems.


*Oral syringes are used in the hospital but they are not available at community pharmacies. Instructing parents is therefore problematic since the measuring devices are different.*



*Syringes used at home should look the same as the ones used in the hospital, differences in scale or color can be confusing.*


### 3.5. Roles and Cooperation of Healthcare Professionals

Based on the focus group discussions, different healthcare professionals have different roles in the drug administration process (Table 5). Physicians prescribe the medication. However, nurses choose the oral dosage form based on their knowledge about every individual child’s ability to take medications. Nurses know different ways of managing drug administration challenges and they inform the physicians on how drug administration may succeed. If problems occur, nurses consult either the clinical pharmacist or the physician at the ward. Clinical pharmacists give advice on the choice of dosage form, drug manipulation and the availability of different dosage forms from the hospital pharmacy.

## 4. Discussion

### 4.1. Main Findings

Healthcare professionals experience several administration challenges related to oral dosage forms, as well as the personal features of the patient, including personality and developmental and health status. Solid oral dosage forms administered to children were connected to swallowing issues caused by the size of the tablet or capsule and a need for modification of the dosage form in order to administer a specific dose, facilitate swallowing or administer the drug through an enteral feeding tube. In this study, typical solutions identified to ease swallowing and mask the taste of the drug included wrapping capsules, whole tablets or pieces of tablets inside a slippery coating (Medcoat^®^) or dispersing crushed tablets, dose powders or the contents of a capsule in juice, juice concentrate or soft food like fruit pure or jam. Mixing medication with food has also been reported in other studies as a method to facilitate swallowing and thus improve acceptability [3,20,23].

In this study, both nurses and pharmacists were concerned about the impact of dosage form manipulations on dose accuracy. Indeed, these concerns are supported by evidence in previous studies. The splitting of tablets into halves or quarters might result in ununiformed segments regarding weight or drug content [24,25,26,27]. A tablet splitter was shown to be a more reliable method to split tablets of different sizes and shapes than the use of a kitchen knife, scissors or manual splitting [28]. Cutting tablets into smaller segments presents a risk for over- or under dosing but the clinical impact is unclear [26,27]. In vitro dissolution studies have shown that orange juice and soft foods such as yoghurt, honey and jam do not significantly affect the drug release of crushed amlodipine, carbamazepine and warfarin compared to the dissolution of whole tablets in water [29]. On the other hand, mixing crushed atenolol with strawberry jam did make the dissolution profile slower. Therefore, avoidance of dosage form manipulation is recommendable if other suitable dosage forms are available.

Healthcare professionals seem to prefer oral liquids and orodispersible tablets as the oral dosage form of choice. Oral liquids were considered to be the most suitable dosage form when administration into a feeding tube is necessary or there is a need for personalized or flexible doses. However, the drug concentration of oral liquids should enable flexible administration of different doses without exceeding a dose volume of a few gulps in order to avoid vomiting and spitting. The manufacturing of different strengths of oral liquids could solve the challenge of administering a suitable dose volume for children of varying ages. Similar results were reported by Alyami et al. who identified oral liquids to be the most preferred dosage form by healthcare professionals, followed by orodispersible tablets, which were also considered to potentially replace the oral liquids [15].

A lack of depot formulations available for children was recognized especially for the treatment of pain and epilepsy. A review article by Trofimiuk et al. listed the potential orally administered modified release formulations for children, of which some are already licensed and others are awaiting license approval [30]. The off-label administration of intravenous drug formulations orally is another situation where the treatment options for pediatric patients is not optimal and a need for dosage form development is obvious. The administration of intravenous formulations orally is common practice in other countries as well [16,31]. In Australia, chewable chocolate-tasting tablets containing midazolam showed better acceptability compared to the commonly used intravenous formulation in a randomized trial involving children receiving premedication before anesthesia [31]. Available formulations and prevailing practices may differ among hospitals and countries. Although there has been an increase in the availability of age-appropriate dosage forms for children, most of these are licensed for use in school-aged children, and the lack of age appropriate dosage forms for the smaller children prevails. The development of pediatric medicines for a relatively small target group is however not financially very attractive for pharmaceutical companies. Ex tempore manufacturing of immediate release formulations is possible on site in hospital pharmacies on-demand but the need for depot formulations is not an easy task to meet.

Investigations into novel oral dosage forms for children have indicated that formulations such as orodispersible mini-tablets or films are promising options in pediatric drug development, enabling personalized and flexible drug administration [7,32]. Dose titrations of enalapril maleate doses in the range of 0.025–4 mg were manageable with orodispersible mini-tablets of two different strengths (0.25 and 1 mg, respectively), facilitating flexible dosing for different patient groups [32]. The production of orodispersible films by means of 2D or 3D printing is considered a possibility for on-demand manufacturing of patient-specific doses [33,34]. Recently, two different printing technologies were used to produce orodispersible films containing four different commonly used pediatric doses of warfarin sodium [7]. It was shown that content uniformity and dose accuracy of the printed orodispersible films were at least equally good as for dose powders currently used for pediatrics. Both orodispersible mini-tablets and films dispersed in water were also suitable for administration through a nasogastric tube without causing blockage of the tube [7,32]. As enteral feeding tubes are widely used in the hospital setting, it is desirable that tests mimicking administration through feeding tubes for children are performed during drug development.

Two themes that were not included in the interview guide emerged from the focus group discussions. One of them was regarding the roles and cooperation of different healthcare professionals and the other one included the challenges related to patient discharge. Physicians, nurses and pharmacists had similar opinions about the roles of different healthcare professionals in the prescribing and administration of drugs. It is evident that when the hospital staff are less experienced with drugs, clinical pharmacists can have a significant role in clarifying and improving drug administration options for children. The way collaboration between all these professionals is organized might also be an important factor, e.g., the clinical pharmacist can give advice about safety and dissolution properties of excipients and the physician could utilize this information at the time of prescribing. Especially if a drug is administered through a feeding tube, the possibility of clogging should be considered. Furthermore, the formulary selection process in the hospital would benefit from multidisciplinary cooperation, including expertise on drug administration practices as well as the safe use of excipients in pediatric patients.

Another phase of care where clinical pharmacists could aid the medication process is patient discharge. Both nurses and clinical pharmacists discussed the deficiency in availability of similar syringes for oral administration in community pharmacies as is used at hospital wards. Marketed pharmaceutical solid formulations containing the same drug substance might differ in their pharmaceutical properties, such as whether the tablet has a break mark or coating, affecting drug manipulations and drug administration. Before patient discharge, nurses teach the child’s caregivers correct drug administration techniques and feeding tube management if needed and the lack of identical drug products and administration devices might cause confusion and a risk of medication errors. Both equipment factors and caregiver education have been identified to have an impact on discharge of pediatric inpatients [35]. Pharmacists at hospitals and in community pharmacies could cooperate to improve the necessary continuum of uniform drug products and administration devices after patient discharge from hospital if an impact on medication safety is apparent.

### 4.2. Strengths and Limitations 

Focus group interviews give information about predefined themes as well as topics emerging from the group discussions. The interview guide was piloted to ensure its functionality. Two investigators coded the transcripts and analyzed the data independently of each other, which improves the reliability of the study. Another strength of this study was that several pediatric subspecialties were represented, although the total amount of participants is usually small in qualitative focus group studies. The study was conducted at one tertiary pediatric hospital in Finland and the results cannot be generalized. However, our findings support previous studies conducted in other countries about oral drug administration to children and bring a holistic view about patient-centered drug administration practices.

The participants in the focus group discussions were healthcare professionals. Therefore, the results in this study reflect the experiences and opinions of healthcare professionals and might differ from the opinions of children themselves or their caregivers. However, the experiences of healthcare professionals caring for pediatric patients at the wards reflect the behaviors and attitudes of children during drug administration and give important information on drug administration challenges related to currently used dosage forms. The opinions of hospitalized pediatric patients and their caregivers on drug administration challenges should further be investigated.

## 5. Conclusions

The focus group discussions revealed both challenges, needs and solutions regarding oral drug administration at hospital wards and information about facilitators to patient discharge. New dosage forms for children of different ages and developmental stages are needed. The dosage forms should be flexible to administer in appropriate doses without modifying the drug product. Furthermore, the size of the solid dosage form or quantity of administered liquid should facilitate swallowing and medication adherence. The possibility of administering the dose through an enteral feeding tube should also be kept in mind. The cooperation of different healthcare professionals in drug prescribing and administration is important, as well as educating the child’s caregivers in correct administration techniques before patient discharge and improving the overall process of patient discharge.

## Figures and Tables

**Figure 1 pharmaceutics-12-00109-f001:**
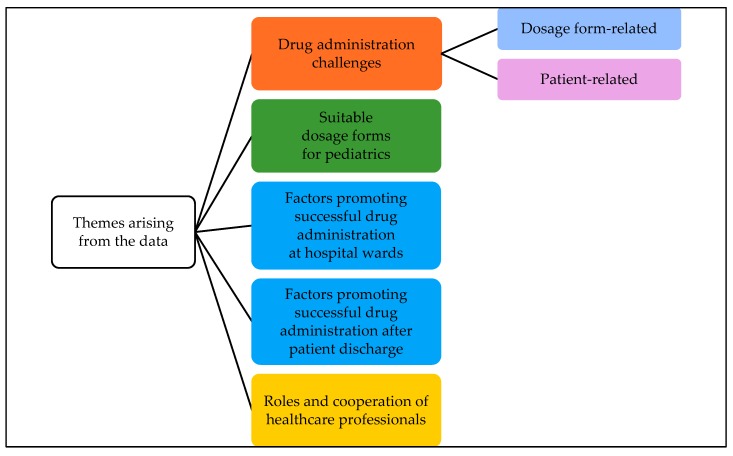
Themes arising from the interviews of healthcare professionals on oral drug administration practices in a children’s hospital grouped into categories and subcategories.

**Figure 2 pharmaceutics-12-00109-f002:**
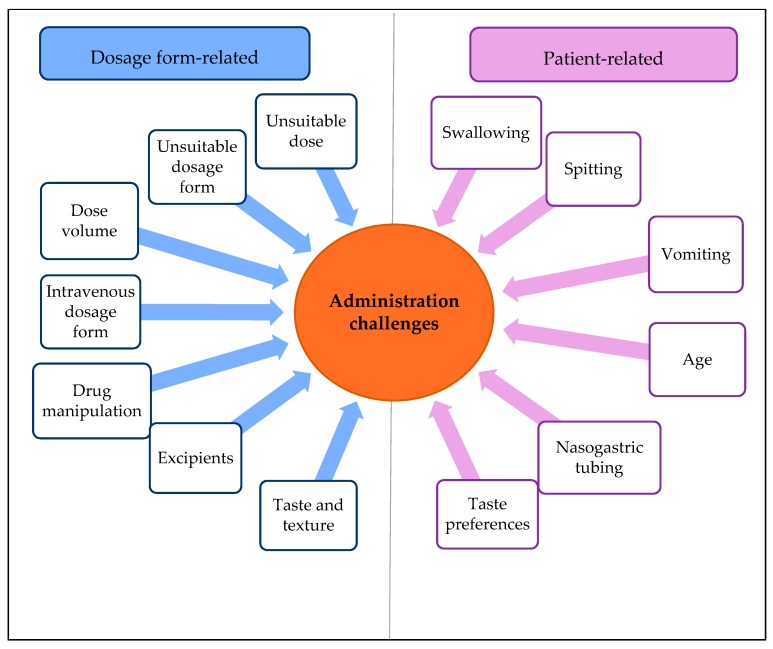
Identified administration challenges of oral medication at pediatric hospital wards divided into dosage form and patient-related challenges.

**Table 1 pharmaceutics-12-00109-t001:** Interview guide for the focus group discussions about oral drug administration practices.

Themes
Suitability of orally administered dosage forms to pediatric patients of different age - benefits - challenges - problems and needs for development
Manipulation of drugs prior administration
Risks associated with pharmaceutical excipients

**Table 2 pharmaceutics-12-00109-t002:** Characteristics of the interviewed healthcare professionals in the focus groups.

Variable	Physicians (n)	Nurses (n)	Pharmacists (n)
Gender			
Female	4	5	6
Male	4	0	0
Total	8	5	6
Age			
20–34	0	1	0
35–49	6	2	5
>50	2	2	1

**Table 3 pharmaceutics-12-00109-t003:** Preferred oral dosage forms for children by healthcare professionals.

Dosage Form	Citations
Orodispersible	*Some children take an orodispersible tablet directly into the mouth, afterwards rinsing with water or milk*
	*You can give orodispersible tablets to babies who eat purees*
	*Orodispersible tablets are of course probably relatively pleasant to take*
Liquid or suspension	*You can administer different doses of oral liquids or suspensions*
	*It is easy to adjust the dose*
	*A liquid is the best alternative for the smallest children*

**Table 4 pharmaceutics-12-00109-t004:** Means to improve medication adherence and successful drug administration.

Professional skills of the nurse	Skillful administration techniques
	Knowledge about useful manipulation methods
	Knowledge about the preferences of each individual child
	Firm guidance
Manipulation of drug products	Cutting a tablet into smaller pieces
	Crushing of tablets
	Covering a bad taste of drug with juice or juice concentrate, glucose solution, water with added lemon concentrate, milk, fruit purees or jam (especially raspberry jam is good if the drug is in small pieces)
	Ex tempore manufacturing of dose powders from commercial drug products
	Dispersing of (crushed) tablets and dose powders before administration
Use of administration aids	Coating (Medcoat^®^) with good taste to cover the drug and facilitate swallowing of tablets, as a whole or in halves or pieces, or capsules.
Consulting different sources of information	Transferring information amongst nurses
	Consulting a clinical pharmacist
	Consulting the hospital pharmacy
	Reading the package information leaflet or summary of product characteristics
Selecting the most appropriate drug product for the child	Ex tempore manufacturing of oral liquids or suspensions
	Choosing the more viscous alternative of two liquid formulations
	Considering the total volume of liquid formulations to be administered
	Using orodispersible tablets and dispersing them with water in a small spoon
	Taking into account possible risks with respect to excipients

**Table 5 pharmaceutics-12-00109-t005:** The roles of different healthcare professionals in prescribing and administering oral medicines at hospital wards.

Profession	Role
Physician	Prescribes the active pharmaceutical ingredient, dose and route of administration. Does not intervene in the choice of dosage form.
	Physician and nurse decide the route of administration together
Nurse	Informs physician which oral dosage form an individual child can take
	Decides which oral dosage form to administer to the child
	Consults physician if problems occur
	Knows what gimmicks to use in drug administration
	Gives feedback to physician about medication administration
Clinical pharmacist	Discusses choice of dosage form with nurse
	Gives advice in drug manipulation and administration matters

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
