# Peer review of "A Focus Group Study about Oral Drug Administration Practices at Hospital Wards—Aspects to Consider in Drug Development of Age-Appropriate Formulations for Children"

_pharmaceutics, 2020, doi:10.3390/pharmaceutics12020109_

Round 1
Reviewer 1 Report
I have enjoyed to read this paper on this relevant topic to pediatric pharmacotherapy.
In the introduction, there is value to add the recently published EAP statement on off label use of drugs in neonates, infants and children (Schiers et al, Eur J Pediatr) to put this aspect in a recent European context.
Along the same line, it may also be interesting to add the EMA guideline on formulations in children to provide the readership access to the statement of the authorities in this ? cfr discussion and the advice to test drugs on ‘adherence’ to plastic tubing.
Propylene glycol ? instead of propylenglycol
There is valuable observational work on the excipient aspect of the clinical pharmacy in pediatric hospital setting in Scandinavia (Valeur K/Horst) that is likely a good illustration of the extent of the issue.
There are two aspects that should be further considered, likely in the discussion part of the paper.
First, the focus group was based on health care providers with different backgrounds. Can the authors speculate or elaborate to what extent their results/assessment/output is either or not different from the opinion of parents and children ?
Second, the focus group had access and experience with the formulations and practices as available within their hospital(s), and these may be different in another setting. To illustrate this, eg
Midazolam might be substituted by buccolam, but rather expensive switch ?
There are different formulations for paracetamol as syrup, with different concentrations, but also with different concentrations on excipients, so how are products ‘selected’ for a given hospital. So is not one of the issues that multidisciplinary groups should ‘predefine’ the criteria for product selection ? How do results compare to other observations of this kind ?
Line 225 (I’m neither native English speaking, but is the word ‘clog’ correct ?)
How is the logistic and legal setting on extemporaneous formulation in Finland ? are the formulations manufactured on site and patient specific, or is manufacturing mainly done by external providers of these products (like UK) ?
Reviewer 2 Report
The manuscript by Rautamo et al. offers an interesting study on the oral drug administration practices at hospital wards, considering several aspects related to the development and the lack of appropriate oral pediatric formulations.
The manuscript is well written, organized and the results clearly presented and discussed.
Suggestions:
This Reviewer would suggest to the Authors to eliminate Table 3 and 5, since a description of the preferred oral dosage forms for children by healthcare professionals and the roles of different healthcare professionals in prescribing and administering oral medicines at hospital wards have been already discussed in paragraphs 3.2 and 3.5 respectively.
